# Vertebro-Vertebral Arteriovenous Fistulae: A Case Series of Endovascular Management at a Single Center

**DOI:** 10.3390/diagnostics14040414

**Published:** 2024-02-13

**Authors:** Pattarawit Withayasuk, Ritthikrai Wichianrat, Boonrerk Sangpetngam, Thaweesak Aurboonyawat, Ekawut Chankaew, Saowanee Homsud, Anchalee Churojana

**Affiliations:** Siriraj Center of Interventional Radiology, Siriraj Hospital, Mahidol University, Bangkok 10700, Thailand; dorawit@gmail.com (P.W.); chonotype@hotmail.com (R.W.); dboonrerk@gmail.com (B.S.); aurboonyawat@gmail.com (T.A.); chankaewek@gmail.com (E.C.); saowanee63999@gmail.com (S.H.)

**Keywords:** vertebro-vertebral arteriovenous fistula (VVF), vertebrovenous fistula, endovascular treatment, neurofibromatosis type 1 (NF-1)

## Abstract

Objective: Vertebro-vertebral arteriovenous fistulae (VVFs) are a rare disorder characterized by a direct shunt between the extracranial vertebral artery and the veins of the vertebral venous plexus. This study aims to comprehensively review the characteristics and outcomes of endovascular treatments for VVFs at our center. Methods: A retrospective review was conducted on 14 patients diagnosed with a VVF who underwent endovascular treatment at Siriraj Hospital from January 2000 to January 2023. The study assessed patient demographics, presentation, fistula location, treatment strategies, endovascular techniques employed, and treatment outcomes. Results: Among the 14 patients, 11 (78.6%) were female, with an age range from 25 to 79 years (median: 50 years). Spontaneous VVFs were observed in 64.3% of the cases, including three associated with neurofibromatosis type 1 (NF-1). Iatrogenic injury accounted for two cases, and three patients had VVFs resulting from traffic accidents. A pulsatile neck mass and tinnitus, with or without neurological deficits, were common presenting symptoms. Active bleeding was observed in three cases with vascular injury, while unilateral proptosis, congestive heart failure, and incidental findings each presented in one patient. All the VVFs were successfully obliterated without major treatment complications. Parent vessel sacrifice was performed in 85.7% of the cases, while vertebral artery preservation was achieved in the remaining two patients. Embolic materials included detachable balloons, detachable coils, and n-butyl cyanoacrylate (NBCA) glue. All the presenting symptoms showed improvement, and no morbidity or mortality was observed. Conclusions: Endovascular embolization is a feasible and effective approach for achieving complete VVF obliteration with safety. Parent artery sacrifice should not be reluctantly performed, particularly when adequate collateral circulation is demonstrated.

## 1. Introduction

A vertebro-vertebral arteriovenous fistula (VVF) manifests through an abnormal direct connection between the extracranial vertebral artery (VA) and the veins constituting the vertebral plexus [1,2,3]. These occurrences, while relatively infrequent, may arise spontaneously or result from mechanical, traumatic, or iatrogenic factors [4,5,6,7,8]. Spontaneous VVFs are associated with conditions such as NF-1, fibromuscular dysplasia, or other connective tissue diseases [3,9,10,11,12,13,14,15,16,17].

The clinical spectrum of VVFs varies from asymptomatic cases to presentations including headaches, pulsatile neck masses, tinnitus, neurological deficits, and, in severe instances, congestive heart failure [1,3,7,18,19,20]. The severity of the symptoms is intricately linked to flow characteristics and the extent of the venous congestion. The primary treatment objective is the complete obliteration of the fistula, with endovascular embolization currently standing as a pivotal and efficacious method for the safe and effective management of these cases [1,15,21,22,23,24,25].

Given the rarity of VVFs as a vascular disorder, limited literature addresses their management. This study aims to contribute to the existing knowledge by conducting a comprehensive review of the characteristics and outcomes associated with the endovascular treatment of VVFs at our center.

## 2. Materials and Methods

A retrospective review was conducted for patients diagnosed with VVFs who underwent endovascular treatment at Siriraj Hospital, Mahidol University, Bangkok, Thailand, spanning from January 2000 to January 2023. Demographic information, such as age, sex, underlying vasculopathy, clinical presentation, and etiology, were recorded. Disease characteristics, including the shunt location, draining-vein pattern, and presence of a venous pouch or an aneurysm, were comprehensively documented.

The locations of the fistulae were precisely defined based on the following vertebral artery segments: V1 segment, from its origin to the transverse foramen of C6; V2 segment, extending from C6 to the transverse foramen of C2; V3 segment, ranging from C2 through C1 to pierce the dura; and V4 segment, representing the intradural segment.

The endovascular treatment strategy was categorized into two main groups: vertebral artery preservation and vertebral artery sacrifice, with a detailed record of the specific embolic materials employed in each group. Angiographic results were further classified as either complete or incomplete VVF occlusion. The success of the treatment was defined by the achievement of the complete obliteration of VVFs, irrespective of the parent artery preservation. Additionally, the study comprehensively assessed treatment complications and patient outcomes.

## 3. Results

There were 14 patients, comprising 3 males and 11 females, with ages ranging from 25 to 79 years and a median age of 49 years (±17.2). Spontaneous cases accounted for 64.3% (9 patients), with 3 cases associated with NF-1 and the remaining 6 patients having no underlying diseases. Among the 5 patients with trauma-related causes, 2 cases were iatrogenic, stemming from complications of central line indwelling and radical neck dissection surgery for external ear canal cancer. The remaining 3 traumatic cases were attributed to traffic accidents.

All the patients underwent successful treatment, achieving complete fistula closure: 12 patients (85.7%) underwent parent artery sacrifice, while 2 patients (14.3%) had parent artery preservation. Notably, all the patients exhibited favorable outcomes. Minor complications, such as transient occipital pain, vertigo, and puncture-site hematoma, were observed. There were no instances of morbidity or mortality.

A detailed comparison of the clinical presentations, fistula locations, specifics of the endovascular treatment, and outcomes between the spontaneous and traumatic VVFs is presented in Table 1. Comprehensive details for each patient are outlined in Table 2.

## 4. Illustrative Cases

Case 1. (Patient No. 8) A 33-year-old female with underlying NF-1 suffered from a left neck mass, tinnitus, paresthesia of the left arm, and cerebellar insufficiency. An MRI (Figure 1a) and a vertebral angiogram (Figure 1b,c) revealed a left high-flow VVF with a large venous pouch. Arterial steal from the right VA (Figure 1d) and left internal carotid artery (ICA) (Figure 1e) was demonstrated. First, an endovascular procedure was performed using two detachable balloons to sacrifice the left VA, which was then accessed from the contralateral VA to occlude the fistula using a few coils to act as the frame, followed by the injection of highly concentrated NBCA (Figure 1f,g). Immediately, the complete obliteration of the VVF was achieved (Figure 1h). The patient had significant clinical improvement at the 3-month follow-up. The right vertebral angiogram demonstrated no evidence of recurrence, with an adequate supply of the posterior circulation (Figure 1i).

Case 2. (Patient No. 6) A 42-year-old male presented with left eye chemosis and proptosis for a few months, with a clinical suspicion of a dural carotid-cavernous fistula (CCF). Bilateral VA access was initiated with the intention to use a detachable balloon for sacrificing the left VA across the fistula while monitoring angiograms from the contralateral VA. The simultaneous injection of the bilateral VA (Figure 2a–c) revealed a spontaneous left VVF with retrograde venous drainage from the vertebral venous plexus to the inferior petrosal sinus, cavernous sinus, and left superior ophthalmic vein, causing symptoms mimicking those of a dural CCF. The first detachable balloon was placed across the fistula; subsequently, the second balloon was detached just beneath the first one. Accidentally, it dislodged from the initial position, resulting in the remaining flow from the VVF. A microcatheter was then navigated via the right VA to select the residual fistula, concluding the treatment with a few detachable coils (Figure 2d). The controlled right VA angiogram (Figure 2e) and left ascending cervical angiogram (Figure 2f) convinced us of the complete fistula closure, with an adequate supply to the posterior circulation.

Case 3. (Patient No. 12) A 73-year-old female, known to have cancer of the right external ear canal, presented with massive bleeding at right posterior neck about 2 weeks after surgery. A contrast-enhanced CT scan (Figure 3a), and a left VA angiogram (Figure 3b) demonstrated an iatrogenic right VVF with a large false aneurysm. Because the parent vertebral artery was hypoplastic, the embolization procedure was conducted from the left VA. After the microcatheter tip was placed just in front of the venous sac, concentrated NBCA was intentionally injected to occlude the aneurysmal neck (Figure 3c,d). The treatment was successful for the parent artery sacrifice. (Figure 3e).

Case 4. (Patient No. 1) A 49-year-old female had a pulsatile neck mass for several years with progressive right arm weakness over a few months. No underlying disease was identified. A CT angiogram (Figure 4a) and right VA angiogram (Figure 4b) revealed a spontaneous VVF with a huge venous pouch at the V3 segment of the right VA. No retrograde supply from the left VA was recognized owing to the hypoplasia of the right V4 segment (Figure 4c). The occlusion of the huge venous sac was achieved first by careful coiling only at the neck of the pouch and subsequently injecting concentrated NBCA (Figure 4d–f). The coil frame was used to assist the proper deposition of the NBCA at the target during the embolization.

Case 5. (Patient No. 5) A 50-year-old female, who had no underlying disease, suffered from progressive right arm weakness. A CT scan (Figure 5a) and a right VA angiogram (Figure 5b) demonstrated a spontaneous VVF at the right V2 segment, at the C4 level, with retrograde venous reflux into peri-medullary veins. The distal right VA was nevertheless accompanying the supply of the basilar artery. No retrograde supply from the left VA (Figure 5c) was visualized, suggesting a low flow in this lower type of nontraumatic VVF. The fistula was obliterated successfully with parent artery sacrifice by placing coils at the site of the fistula and a detachable balloon proximally via right VA access (Figure 5d–f) The neurological symptoms consequently subsided.

Case 6. (Patient No. 3) A 79-year-old female, who was being treated for congestive heart failure, was discovered as having a spontaneous VVF at the V1 segment of the right VA connecting to the right internal jugular vein according to a CT angiogram (Figure 6a,b). The proximal right VA showed obvious dilatation. An embolization was planned to occlude the primary vein of the fistula with the preservation of the parent artery using detachable coils (Figure 6c–f). After finishing the procedure, a residual shunt with a low flow to the internal jugular vein was observed (Figure 6g). The congestive heart failure has significantly improved since then. At the 1-month follow-up, the CT angiogram (Figure 6h) further demonstrated the complete obliteration of the fistula, with the normal appearance of the affected vertebral artery.

Case 7. (Patient No. 13) A 57-year-old man presented with a right pulsatile neck mass and right arm weakness 2 weeks after a traffic accident. A CT angiogram (Figure 7a) and left VA (Figure 7b) and right VA (Figure 7c) angiograms demonstrated a fistula at the V2 segment of the right VA, with a small venous pouch and obvious total steal of the right VA flow. The preservation of the parent artery was initially planned using detachable coils placed at the proximal venous pouch and fistula site. Nevertheless, an indention on the parent artery with near total occlusion was recognized (Figure 7d,e). The posterior circulation was solely supplied from the left VA (Figure 7f). Thus, heparin was administered for a few days after the procedure to prevent a thromboembolism. Subsequently, the right VA was further occluded on follow-up, without any relevant symptoms.

## 5. Discussion

VVFs are a rare neurovascular disease, and owing to their infrequent clinical manifestation, a comprehensive understanding of the associated morbidity and mortality remains underexplored. However, untreated symptomatic cases pose the potential for heightened adverse outcomes [1,3,26]. Approximately one-third of VVF cases are spontaneous, often observed in individuals with underlying genetic disorders, such as NF-1, or connective-tissue disorders, like Ehlers–Danlos syndrome or fibromuscular dysplasia [10,11,12,13,14,16,17].

A study conducted by Aljobeh A. et al. represents the most extensive review of VVFs to date, encompassing 128 case reports and 16 case series, totaling 280 patients. Their findings revealed that spontaneous VVFs accounted for 49% of the cases, predominantly affecting young women and commonly located between C1 and C2. Additionally, a noteworthy association with NF-1 was observed in 25% of the cases [1].

In our study, spontaneous causes were more prevalent in VVFs compared to traumatic etiologies (64.3% vs. 35.7%). Notably, 21.4% of the spontaneous VVFs were associated with NF-1 (Figure 1), while the remaining 42.9% lacked discernible underlying causes. Among traumatic VVFs, 21.4% resulted from traffic accidents, and 14.3% were iatrogenic. Intriguingly, one iatrogenic case stemmed from a complication during the central line indwelling procedure at the right jugular vein (Patient No. 2).

When contrasting our findings with those of Aljobeh A. et al., similarities in demographic patterns and spontaneous VVF characteristics were noted. However, our study revealed a slightly higher proportion of spontaneous cases, and the patients exhibited an older mean age with varying ages at presentation.

VVF symptoms predominantly emanated from a high-flow shunt connecting the vertebral artery to the adjacent venous plexus. Neurological manifestations resulted from venous congestion or retrograde intradural venous reflux. A large venous pouch exerted a massive effect, generating compressive symptoms on nerve roots, while a substantial high-flow shunt could lead to congestive heart failure.

Consistent with the existing literature, the primary presenting symptom was a bruit, observed in around 70% of the patients, while extremity weakness or numbness occurred in up to 20% of the cases [1,18]. Tinnitus, as described by Beaujeux R.L. et al., was similarly prevalent in 46.7% of the cases [2].

In our series, prevalent clinical presentations encompassed neck masses and tinnitus, often co-occurring in a single patient. Notably, neurological deficits were observed in 35.7% of the patients, manifesting as arm weakness, radicular pain, arm paresthesia, and cerebellar insufficiency. All the instances of neurological deficits were associated with intradural venous reflux. In the unique case of a spontaneous VVF with proptosis, retrograde venous reflux into the inferior petrosal sinus and extending into the cavernous sinus and superior ophthalmic vein prompted a referral under the initial impression of a dural carotid-cavernous fistula (Figure 2). A similar observation was reported by Kobkitsuksakul et al., highlighting the challenging diagnostic differentiation between VVFs and carotid-cavernous fistulae [27].

The literature has underscored the significance of hemorrhagic complications as presentations in VVF cases. Tomoo I. et al. documented the compelling case of an acute subarachnoid hemorrhage from a VVF at the left V1 vertebral artery segment draining into the peri-medullary veins. Despite the initial treatment refusal, the patient subsequently suffered from an intramedullary hemorrhage, leading to an unfavorable clinical outcome [28]. Similarly, Brian P. et al. reported a VVF at the V2 segment of the vertebral artery with retrograde venous reflux into intracranial veins, presenting with an acute subarachnoid hemorrhage and an intraventricular hemorrhage. This patient experienced symptom improvement after treatment [29]. In our study, three cases presented with extra-cranial hemorrhages; all were attributed to traumatic causes. One patient, who had active bleeding from the right VVF with a large false aneurysm, underwent neck surgery for cervical lymph node dissection (Figure 3). Another case was iatrogenic, linked to central line indwelling (Patient No. 2), and the third resulted from a traffic accident, revealing an epidural hematoma during the neck exploration (Patient No. 14). Importantly, no spontaneous VVFs in our experience were associated with hemorrhages. This collective evidence emphasizes the diverse clinical presentations of VVFs and underscores the importance of tailored management strategies based on the underlying etiology and hemodynamic characteristics of each case.

The predominant location of the spontaneous VVFs in our study was at the V3 segment of the vertebral artery, constituting 66.7% (6/9) of the cases. This observation aligns with the prevailing distribution of spontaneous VVFs, often manifesting at the V3 segment [1]. Conversely, traumatic cases did not exhibit a specific pattern in the fistula’s location, with occurrences at the V3, V2, and V1 segments, corresponding to the site of the injury. Intriguingly, all three cases of spontaneous VVFs associated with NF-1 were localized at these specific locations. This distribution may be linked to the physiological movement of the vertebral artery during head rotations, where the transition from V2 to V3 after exiting the transverse foramina at the C1 and C2 levels renders one side susceptible to compression during ipsilateral head turns [30,31].

Lasjaunias P. et al. provided a valuable classification of nontraumatic VVFs into upper and lower types, shedding light on variations in shunt locations. The upper type, situated at the C1 and C2 levels or V3 segment of the vertebral artery, was associated with a high-flow pathology and commonly observed in childhood. In contrast, the lower type, found at the C3 to C7 levels (V1 and V2 segments of the vertebral artery), typically displayed a low-flow pathology and presented in young adults [32].

In our series, patients with spontaneous VVFs at the V3 segment exhibited a broad age range, spanning from 27 to 70 years at presentation. Notably, 66.7% (4 cases) exhibited a large venous pouch, indicative of a high-flow stage (Figure 4). Conversely, three spontaneous VVFs of the lower type showed no large venous pouches, despite two of them draining into the vertebral venous plexus and peri-medullary veins (Figure 5). This distinction may be attributed to the relatively lower flow in the lower type compared with the upper type, as proposed by Lasjaunias P. et al. [32].

In formulating our treatment strategy, we meticulously considered endovascular intervention for all the cases on an elective basis, with the exception of two cases (Patients No. 12 and 14) that presented with active bleeding.

Prior to treatment planning, a comprehensive evaluation encompassed not only the precise location of the fistula but also an exhaustive assessment of the complete cerebral circulation. This evaluation extended to the arteries supplying the posterior fossa, inclusive of the anastomosis in the upper cervical region.

Our treatment strategy sought to achieve an effective fistula occlusion concomitant with the sacrifice of the parent vertebral artery. The preservation of the vertebral artery was contemplated under the following specific circumstances:The affected VA held dominance in the vascular supply of the posterior circulation;The shunt presented as being small without the presence of a sizable venous pouch.

Gratifyingly, in all the cases where sacrifice was considered, the contralateral VA demonstrated patency with ample flow to supply the basilar artery system. Additionally, certain cases had already manifested evidence of a steal phenomenon. This meticulous approach ensured a comprehensive and individualized treatment plan for optimal outcomes in each scenario.

Within the context of vertebral artery sacrifice at our center, our primary consideration for embolic devices leaned toward the use of a detachable balloon. The procedural sequence involved approaching the affected VA and detaching the balloon across the fistula. In certain scenarios, a second balloon, positioned just below the first, served as a safety measure if the detached balloon’s stability was in question. However, challenges arose, particularly in instances where the shunt exhibited a significant size, high-flow characteristics, and a sump effect from the contralateral VA, especially with a large venous pouch. These factors impose limitations on passing the balloon across the neck of the pouch. Consequently, an alternative approach through the contralateral vertebral artery was adopted. This involved navigating a microcatheter into the venous sac, facilitating the occlusion of the fistula using n-butyl cyanoacrylate (NBCA) glue and/or coils in conjunction with utilizing a balloon for the sacrifice in the ipsilateral VA (see illustrative case 1).

In a specific case within our series, a traumatic VVF involving a hypoplastic vertebral artery, the balloon sacrifice procedure was deemed unfeasible. Instead, an embolization technique was employed by superselecting via the contralateral VA, crossing the vertebrobasilar junction to reach the neck of the false aneurysm. The injection of a high concentration of NBCA successfully closed the shunt along with sacrificing the vessel (see illustrative case 3).

Notably, despite the presence of a large venous pouch in some VVF cases, retrograde supply from the contralateral VA was absent, usually owing to the hypoplasia of the ipsilateral V4 segment. This allowed for vertebral artery sacrifice solely through coils and/or NBCA embolization, without the necessity for using a detachable balloon (see illustrative case 4).

It is imperative to acknowledge a disadvantage associated with the use of detachable balloons, specifically, the potential for dislodgement from the proper position during the detachment (see illustrative case 2).

In cases involving a substantial venous pouch or false aneurysm, our treatment paradigm emphasized the disconnection at the neck of the pouch or shunting zone. It was unnecessary to entirely fill the sac with coils or liquid embolic materials.

The preservation of the parent artery was usually planned when the fistula presented without a significant size and lacked a large venous pouch. In such scenarios, our embolization technique honed in on the fistula and proximal vein, with detachable coils often emerging as the preferred embolic material. This involved the placement of a microcatheter’s tip beyond the parent artery, strategically aimed at occluding the primary draining vein (Figure 6).

However, in instances where the intention was to preserve the affected vertebral artery, the conclusion of the fistula obliteration might inadvertently indent the parent artery. In such cases, a continuous heparin regimen for a few days was recommended to mitigate the risk of a further thromboembolism, especially when parent artery sacrifice was not the desired outcome (Figure 7).

Our strategic treatment approach yielded a remarkable success rate, with all the cases undergoing successful endovascular treatment in a single session. Parent artery sacrifice accounted for 85.7% of the cases, while 14.3% involved parent artery preservation. Noteworthy is the fact that the two cases where the vertebral artery could be preserved belonged to the spontaneous group. In contrast, all the traumatic VVFs necessitated parent artery sacrifice, even in cases where the parent artery was initially preserved but subsequently indented by the coil mesh. Despite heparin administration, progressive occlusion of the parent artery ensued (see illustrative case 7).

Significantly, the successful treatment in 57% of the cases (8 patients) was achieved without deploying a detachable balloon. This included the use of only coils, NBCA, or a combination thereof. Particularly, parent arteries were preserved in VVFs treated solely with coils. In the remaining 43% of the cases (6 patients) where a balloon accompanied the embolization, parent arteries were sacrificed, with three cases exclusively utilizing balloons.

Crucially, our endovascular approach resulted in a lack of morbidity and mortality. All the patients, except for one who presented asymptomatically with an incidental finding, experienced notable symptom improvement, and no instances of VVF recurrence were observed.

Our study aligns with the comprehensive review conducted by Aljobeh A. et al., who delineated diverse treatment modalities for vertebral arteriovenous fistulae (VVFs). Notably, the majority of cases, approximately 73.1%, underwent endovascular deconstruction, characterized by the occlusion of the fistula and its feeding vessels. Endovascular construction, preserving the parent artery through the use of detachable balloons and/or coils (with or without stent supports), was applied in 11.1% of the cases. Surgical intervention and a no-treatment approach accounted for 9.2% and 6.6%, respectively. The overall success rate of the treatment in their study reached 90%, with treatment-related permanent morbidity at 3.3% and mortality at 1.5% [1]. Of particular note is our study’s reporting of a more favorable success rate in endovascular treatment and overall patient outcomes. Remarkably, none of our patients underwent surgery for the treatment of a VVF.

Concerning detachable balloons, although proven to be safe and facile for arterial occlusion, their availability remains a concern in numerous countries, particularly in Europe and the United States [3,22,33,34]. Detachable coils, although serving as an alternative for parent artery sacrifice, present drawbacks, such as a higher cost and a prolonged procedural time.

To address these limitations, there have been reports for employing a combination of distal coils and proximal Amplatzer™ Vascular Plugs or the Woven EndoBridge (WEB™)-device for vertebral artery sacrifice [35,36]. However, it is pertinent to note that our study did not incorporate experiences with these devices.

Transarterial coil embolization, a versatile technique employed for both parent artery sacrifice and preservation, emerges as a prominent and straightforward method [3,20,22,37,38,39,40,41,42]. The landscape of endovascular treatment options has witnessed significant expansion, with documented approaches including transvenous strategies for coiling the venous outlets of fistulae and the utilization of dimethyl sulfoxide (DMSO)-based embolic agents, such as Onyx, Squid, and PHIL. Direct vertebral artery puncture was also reported in the case of a recurrent VVF post-surgical trapping. The procedure was successfully performed under fluoroscopic guidance, followed by microcatheter superselection and 20% NBCA embolization [20,21,40,43,44,45]. A procedural complication involving Onyx migration to the origin of the posterior inferior cerebellar artery (PICA) was also reported; however, successful management was achieved through aspiration [21].

At our institute, NBCA stands as the exclusive liquid embolic material employed in VVF treatment, chosen for its efficacy in managing high-flow shunts owing to its rapid polymerization. NBCA, or cyanoacrylate glue, is a potent adhesive agent with a low viscosity and rapidly hardens upon contact with blood [46,47]. Prior to its administration, NBCA is mixed with ethiodized oil (Lipiodol) to enhance radiopacity and delay polymerization. The concentration of the NBCA is individually adjusted based on the flow pattern in each VVF, with higher shunt flows requiring higher NBCA concentration. When injecting NBCA into the venous pouch, meticulous care is taken to ensure a sufficient safety margin, preventing reflux into the distal vertebral artery or anastomosis. Procedural complications during glue injections are associated with nontargeted embolization to more distal venous drainage, reflux to more proximal arteries, or spillage to anastomoses.

The use of NBCA embolization poses challenges, particularly in less experienced settings. Concerns about potential discomfort have prompted the consideration of alternative approaches, such as the combination of coils and DMSO-based embolic agents. However, cost-effectiveness considerations highlight that these alternative embolic agents are considerably more expensive than NBCA.

The imperative to preserve the affected VA became evident in cases where the contralateral VA exhibited diminution or termination into the posterior inferior cerebellar artery (PICA). When preservation proved to be unattainable, resorting to surgical trapping with a bypass emerged as a feasible alternative treatment. Reports have also indicated the utilization of stent graft deployment across fistulae for treating traumatic VVFs, although complete occlusion was not immediately achieved [24,48,49,50]. Fortunately, in our series, no such anatomical obstacles were identified.

Complications arising from endovascular treatment may manifest as reperfusion hemorrhages, attributable to the abrupt closure of the fistulae redirecting the flow immediately to the intracranial circulation [1]. Conversely, ischemic complications resulting from thromboembolisms may manifest in patients undergoing parent artery sacrifice [1]. Special caution is warranted in patients with spontaneous fistulae, as they may have underlying connective-tissue abnormalities predisposing them to injury. Notably, none of our patients experienced reperfusion hemorrhages or ischemic complications.

Although the endovascular treatment of VVFs proves to be feasible and effective, the crux lies in the critical diagnosis of the disease, particularly if presentations extend beyond a neck mass. The elusive nature of the disease may lead to oversight by general practitioners, emphasizing the need for vigilance in differential diagnoses. The request for imaging studies typically excludes the neck or cervical spine region unless the possibility of a VVF is actively considered.

The limitation of our study lies in the rarity of the disease, posing challenges in recruiting a substantial number of cases.

## 6. Conclusions

VVFs stand as a rare neurovascular ailment, with occurrences either spontaneous or resultant from vascular injury.

In our series of 14 patients, during a period of 23 years, we demonstrated the efficacy of endovascular treatment for VVFs, yielding promising patient outcomes. In terms of endovascular embolization, we consider that parent artery sacrifice should not be hesitated upon, provided that adequate collateral circulation of the basilar artery is demonstrated. Detachable balloons, coils, and n-butyl cyanoacrylate (NBCA) glue emerge as effective embolic materials for the closure of fistula. However, our results may be constrained and not universally applicable to diverse regions owing to the limited number of patients in the study.

## Figures and Tables

**Figure 1 diagnostics-14-00414-f001:**
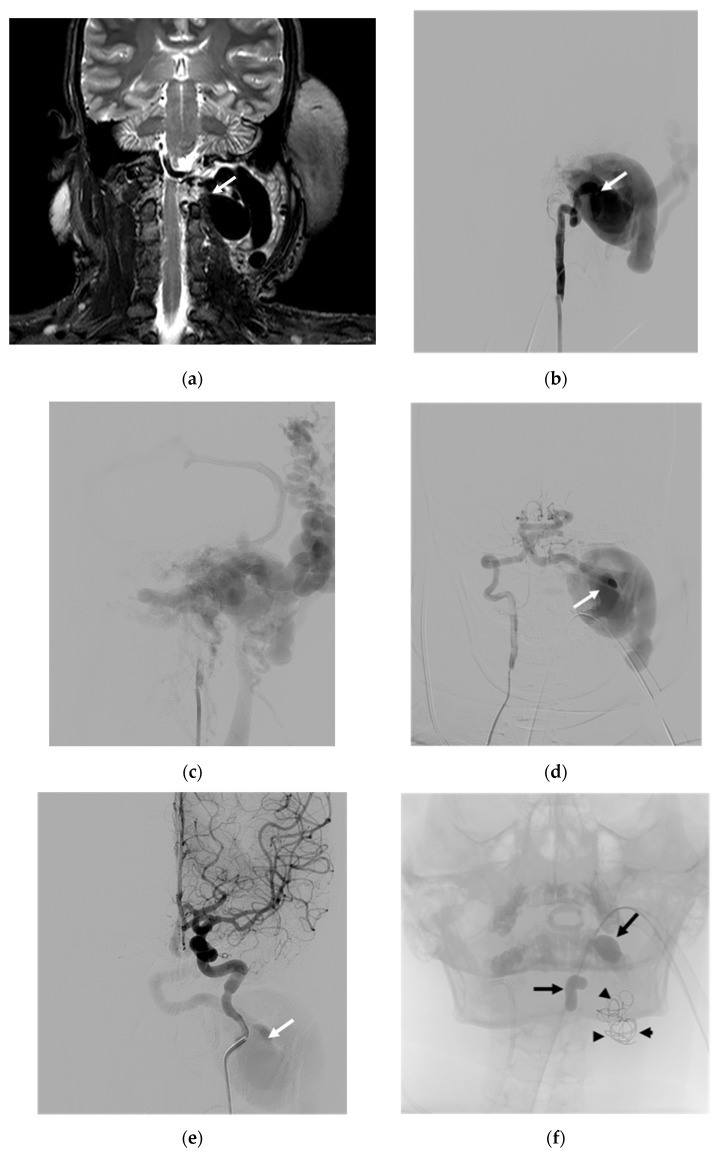
Illustrative case 1. (**a**) Coronal T2W MRI and (**b**) early and (**c**) late arterial phases of the frontal projection of the left VA angiogram, showing a spontaneous left VVF with a large venous pouch. (**d**) Right VA and (**e**) right ICA angiograms demonstrating arterial steal. (**f**) Scout image showing two balloons detaching for the sacrifice of the left VA and a few coils deploying in the venous pouch via right VA approach. (**g**) Superselective injection of NBCA at the aneurysmal neck. (**h**) Controlled right VA angiogram revealing the disappearance of the VVF. (**i**) The follow-up study at 3 months demonstrating no evidence of recurrence. White arrows: venous pouch; black arrows: detached balloon; arrowheads: coil mesh; dashed arrow: NBCA cast.

**Figure 2 diagnostics-14-00414-f002:**
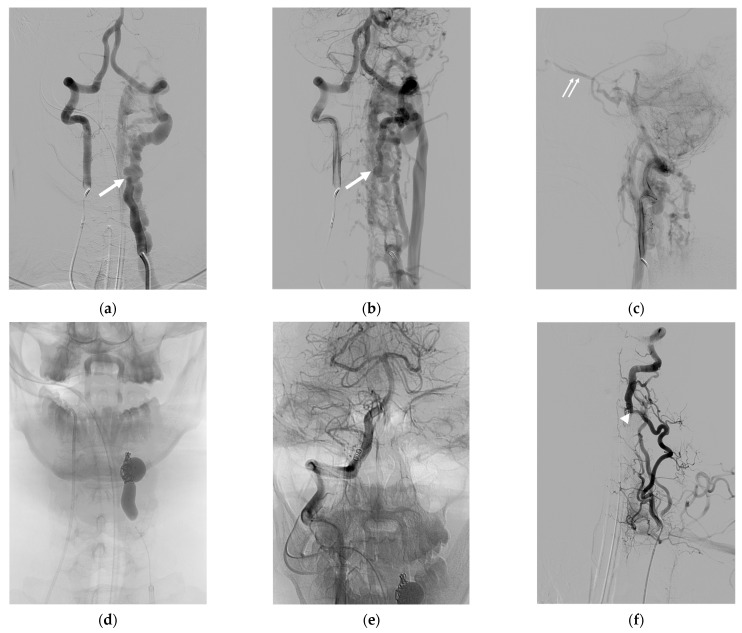
Illustrative case 2. (**a**,**b**) AP and (**c**) lateral projections of the simultaneous injection of bilateral vertebral arteries, revealing a spontaneous VVF at the V2 segment of the left VA, with retrograde venous drainage from the vertebral venous plexus to the inferior petrosal sinus, cavernous sinus, and left superior ophthalmic vein. (**d**) Successful obliteration of the shunt with the sacrifice of the parent artery using detachable balloons and coils. (**e**) Controlled right VA angiogram showing successful VVF closure, with an adequate supply to the posterior circulation. (**f**) Left ascending cervical artery injection demonstrating the preservation of the distal left vertebral artery flow from preexisting anastomosis at the C4 level. White arrows: fistula site; small double arrows: superior ophthalmic vein; white arrowhead: anastomotic site.

**Figure 3 diagnostics-14-00414-f003:**
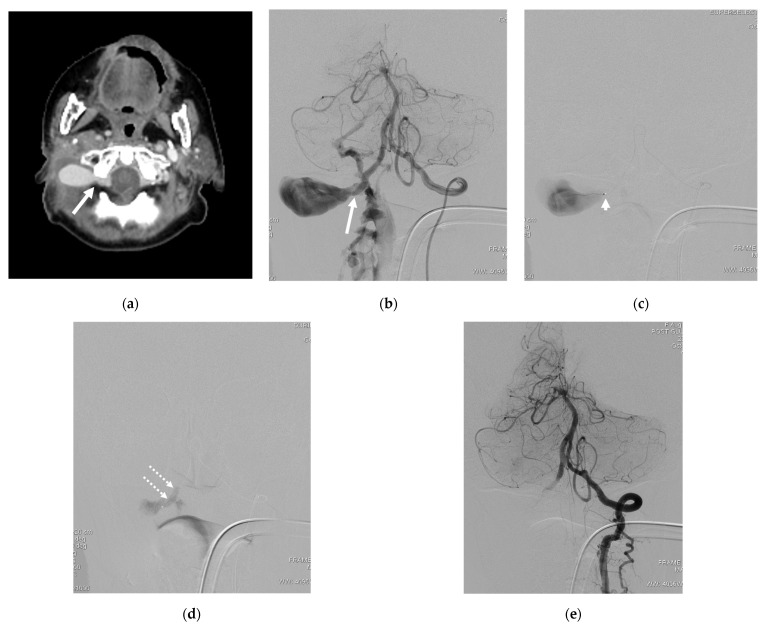
Illustrative case 3. (**a**) Contrast-enhanced CT scan, axial view, and (**b**) left VA angiogram, revealing a right VVF at the V3 segment with a large false aneurysm. (**c**) Superselective catheterization from the left VA with the microcatheter tip placed just in front of the venous sac. (**d**) NBCA cast deposition only at the aneurysmal neck. (**e**) Angiogram of the left VA post embolization, revealing the complete obliteration of the VVF and aneurysm while preserving the distal flow. White arrows: fistula site; arrowhead: microcatheter tip; double dashed arrows: NBCA cast.

**Figure 4 diagnostics-14-00414-f004:**
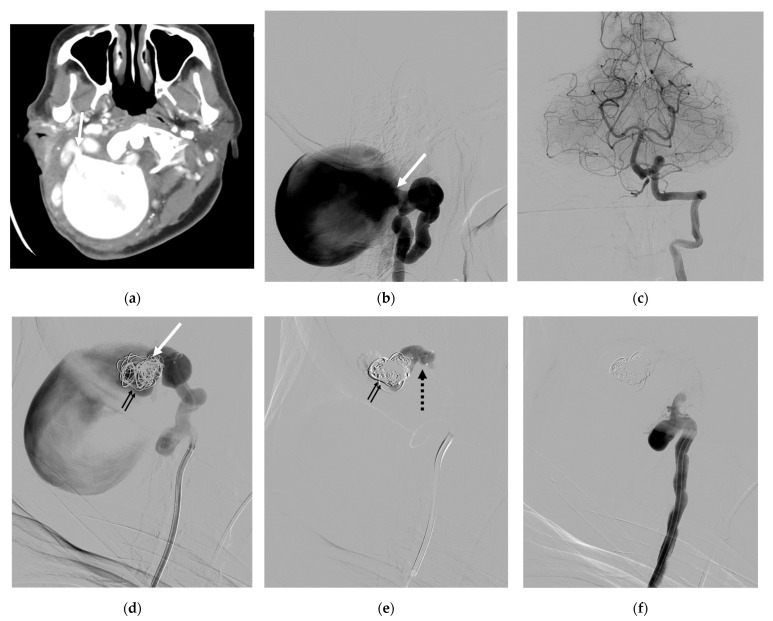
Illustrative case 4. (**a**) CT angiogram, axial view, and (**b**) the lateral projection of the right VA angiogram, demonstrating a high-flow VVF with a huge venous pouch at the V3 segment of the right VA. (**c**) Left VA angiogram revealing the sole supply of the basilar system and the hypoplasia of the right V4 segment, preventing retrograde feeding to the shunt. (**d**) Coils and (**e**) NBCA deposition at the neck of the venous pouch, resulting in the complete disconnection of the venous pouch (**f**). White arrows: aneurysmal neck; small double black arrows: coil mesh; black dashed arrow: NBCA cast.

**Figure 5 diagnostics-14-00414-f005:**
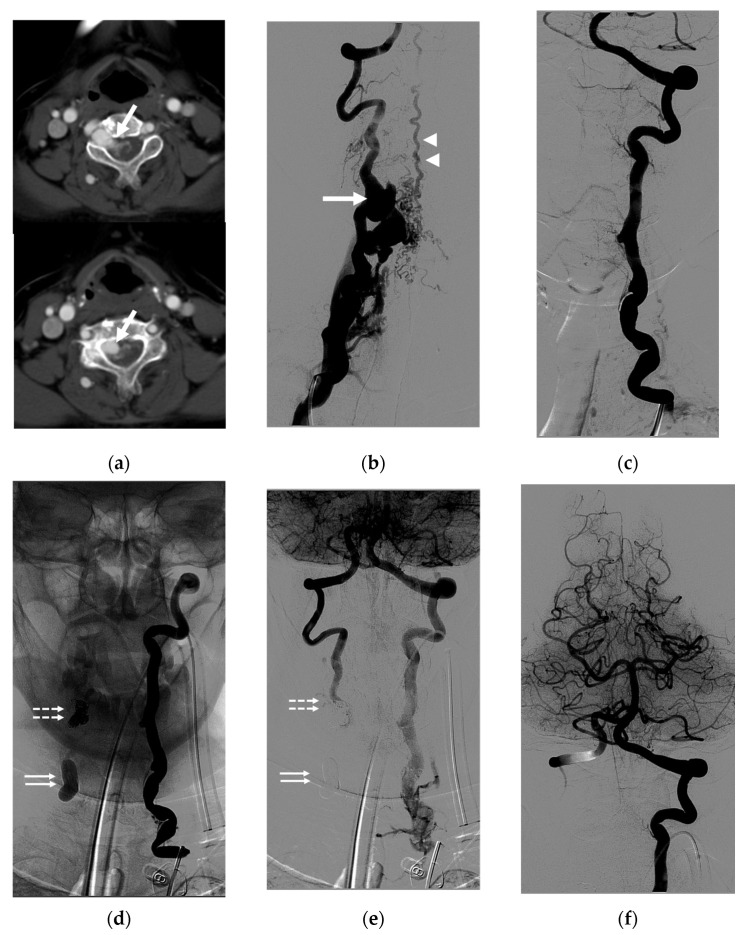
Illustrative case 5. (**a**) Axial views of the CT scan and (**b**) frontal projection of the right VA angiogram, demonstrating a VVF at the right V2 segment, at the C4 level, with retrograde venous reflux into peri-medullary veins, indicating a lack of evidence for arterial steal and a large venous pouch. (**c**) Left VA angiogram showing no supply to the shunt. (**d**–**f**) Controlled left VA angiogram revealing the complete obliteration of the VVF with parent artery sacrifice using coils and NBCA. Note the retrograde filling of the right VA and adequate supply to the posterior circulation. White arrows: fistula site; white arrowheads: peri-medullary vein; double dashed arrows: coil mesh; double white arrows: detached balloon.

**Figure 6 diagnostics-14-00414-f006:**
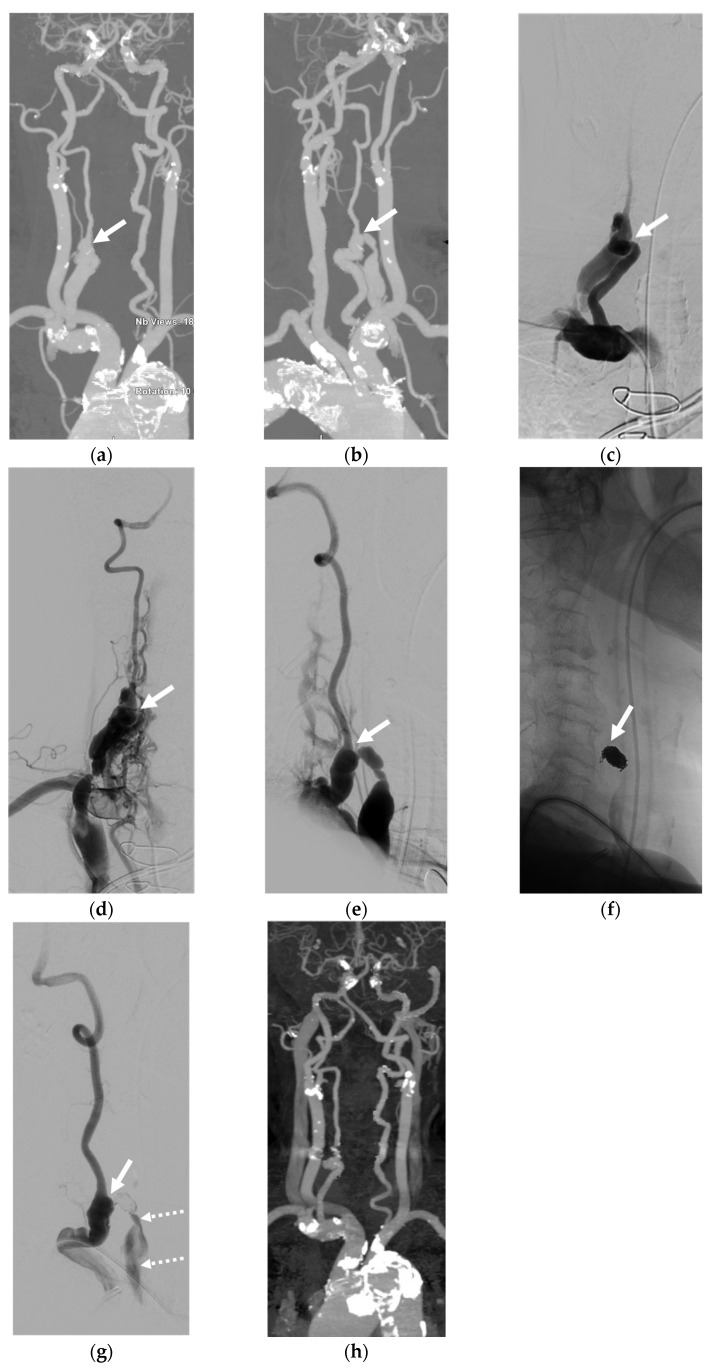
Illustrative case 6. (**a**) AP and (**b**) oblique views of maximal intensity projection images of the CT angiogram. (**c**,**d**) Frontal and (**e**) lateral projections of the right vertebral angiogram, revealing a spontaneous VVF at the V1 segment of the right VA connecting to the right internal jugular vein (IJV). (**f**) Lateral view of the scout image showing the coil mesh. (**g**) Controlled right VA angiogram in the lateral view, demonstrating an immediate low residual flow to the IJV. (**h**) CT angiogram follow-up at 1 month, revealing the complete obliteration of the VVF and decreased size of the proximal VA. White arrows: fistula site; dashed arrows: residual shunt.

**Figure 7 diagnostics-14-00414-f007:**
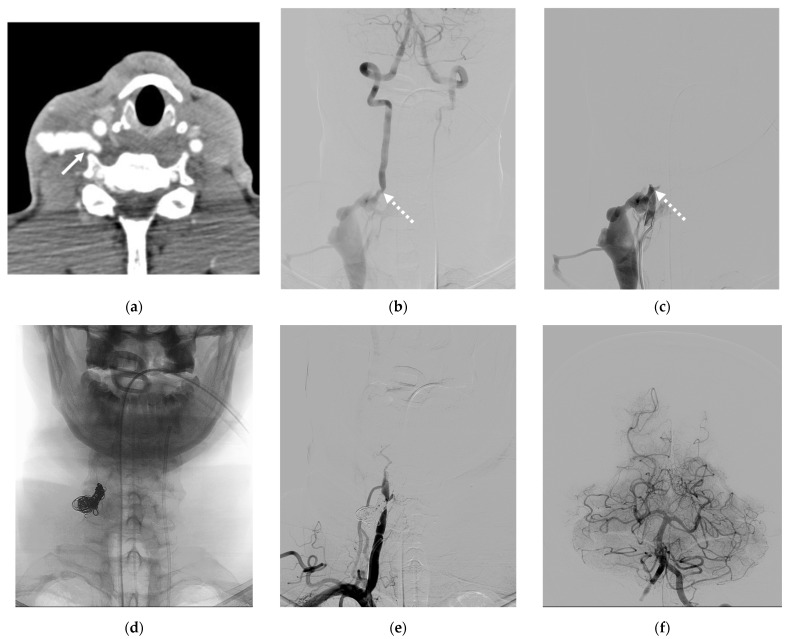
Illustrative case 7. (**a**) CT angiogram, axial view, and (**b**) left VA and (**c**) right VA angiograms revealing a fistula at the V2 segment of the right VA with a small venous pouch draining into the right IJV. Note the total steal of the supply from the left VA. (**d**) Coil mesh at the fistula site and venous pouch. (**e**) Controlled right VA angiogram showing the immediate and complete obliteration of the VVF with near total occlusion of the parent artery. (**f**) Left VA angiogram post embolization, demonstrating adequate supply to the posterior fossa. White arrow: venous pouch; white dashed arrows: fistula site.

**Table 1 diagnostics-14-00414-t001:** A comparison between spontaneous and traumatic VVFs.

Variables	Number of Spontaneous VVFs (%)	Number of Traumatic VVFs (%)	Total Number of VVFs (%)
Number of patients	9 (64.3%)	5 (35.7%)	14 (100%)
Sex (male or female)	1:8	2:3	3:11
Age (years), median ± SD	49 ± 17.9	57 ± 15.7	49 ± 17.2
Symptoms			
Tinnitus	5	0	5 (35.7%)
Neck mass	3	1	4 (28.6%)
Arm weakness	3	1	4 (28.6%)
Radicular pain	1	0	1 (7.1%)
Cerebellar insufficiency	1	0	1 (7.1%)
Proptosis	1	0	1 (7.1%)
Congestive heart failure	1	0	1 (7.1%)
Bleeding	0	3	3 (21.4%)
Incidental finding	1	0	1 (7.1%)
Shunt locations			
V3 segment	6	2	8 (57.1%)
C1 level	3	1	4 (28.6%)
C2 level	3	1	4 (28.6%)
V2 segment	2	2	4 (28.6%)
C4 level	2	0	2 (14.3%)
C4–5 levels	0	1	1 (7.1%)
C5–6 levels	0	1	1 (7.1%)
V1 segment	1	1	2 (14.3%)
C6–7 levels	1	0	1 (7.1%)
T1 level	0	1	1 (7.1%)
Venous pouch/Venous aneurysm	4	2	6 (42.9%)
Embolization techniques			
Parent vessel sacrifice	7	5	12 (85.7%)
Coil alone	0	2	2 (14.3%)
Balloon alone	1	2	3 (21.4%)
NBCA alone	1	1	2 (14.3%)
Coil + balloon	2	0	2 (14.3%)
Coil + NBCA	2	0	2 (14.3%)
Coil + balloon + NBCA	1	0	1 (7.1%)
Parent vessel preservation	2	0	2 (14.3%)
Coil alone	2	0	2 (14.3%)
Clinical outcome			
Improved	8	5	13 (92.9%)
Stable (no symptoms)	1	0	1 (7.1%)
Complications	5	0	5 (35.7%)
Transient occipital pain	3	0	3 (21.4%)
Puncture-site hematoma	1	0	1 (7.1%)
Transient vertigo	1	0	1 (7.1%)

**Table 2 diagnostics-14-00414-t002:** Details of each patient.

Pt. No.	Sex	Age	Symptoms	Etiology	Type	Pattern of Venous Drainage	Venous Pouch	Embolic Materials	Outcome of VA	Clinical Outcome	Complications
1	F	49	Neck mass; tinnitus; right arm weakness	Unknown	S	Radicular vein; VVP with venous pouch	Y	C and NBCA	O	Improved	Transient occipital pain
2	F	33	Bleeding	Central line injury	T	VVP	N	C	O	Improved	None
3	F	79	Congestive heart failure	Unknown	S	IJV; subclavian vein	N	C	P	Improved	Puncture-site hematoma
4	F	62	Tinnitus	Unknown	S	VVP	N	B	O	Improved	Transient occipital pain
5	F	50	Right arm weakness	Unknown	S	Radicular vein; peri- medullary veins	N	B and C	O	Improved	None
6	M	42	Proptosis of left eye	Unknown	S	Epidural veins; VVP to IPS; CS and left SOV	N	B and C	O	Improved	None
7	F	70	Radicular pain; right arm weakness; tinnitus	Unknown	S	Lateral epidural vein	N	C	P	Stable	None
8	F	33	Neck mass; tinnitus; paresthesia of left arm; ataxia	NF-1	S	IJV; VVP with venous pouch	Y	B, C, and NBCA	O	Improved	Vertigo
9	F	25	Neck mass	NF-1	S	VVP with venous pouch	Y	C and NBCA	O	Improved	None
10	F	27	Incidental finding	NF-1	S	Lateral epidural vein with venous pouch	Y	NBCA	O	Improved	None
11	F	64	Neck mass	Traffic accident	T	VVP with venous pouch	N	B	O	Improved	Transient occipital pain
12	F	73	Bleeding	Post-surgery	T	VVP with venous pouch	Y	NBCA	O	Improved	None
13	M	57	Neck mass; right arm weakness	Traffic accident	T	Radicular vein; IJV with venous pouch	Y	C	O	Improved	None
14	M	36	Bleeding in epidural space after surgery	Traffic accident	T	VVP with ruptured venous pouch in epidural space	N	B	O	Improved	None

Abbreviations: VA = vertebral artery; F = female; M = male; NF-1 = neurofibromatosis type 1; S = spontaneous type; T = traumatic type; O = occlusion; P = preservation; VVP = vertebral venous plexus; IJV = internal jugular vein; IPS = inferior petrosal sinus; CS = cavernous sinus; SOV = superior ophthalmic vein; Y = yes; N = No; NBCA = n-butyl cyanoacrylate glue; C= coil; B = detachable balloon.

## Data Availability

All data are available from the corresponding author upon reasonable request.

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
