# Peer review of "Vertebro-Vertebral Arteriovenous Fistulae: A Case Series of Endovascular Management at a Single Center"

_diagnostics, 2024, doi:10.3390/diagnostics14040414_

Round 1

Reviewer 1 Report

Comments and Suggestions for Authors

This study aims to review the characteristics and results of endovascular treatments for VVF.

The study is of a rare pathology, and with a very small sample of 14 patients.

To be published, it is necessary to change the title of the article and call it a pilot study, or case study, as it gives rise to a misinterpretation.

As for the methodology, it is a description of cases, which can provide knowledge to experts on the subject, on specific occasions.

Regarding the changes, in addition to the title, much more humility is suggested, since it is a study very limited by the sample, without statistics, so the results, and conclusions, are limited.

You cannot say, as in the summary, the word exhaustive, and expressions that are too arrogant, for such a limited study.

The presentation of the tables would be appreciated if it were simpler and more understandable.

Author Response

I correct the  title, re-edit the table 1 and table 2 to more understandable and some grammar correction. Please see the attachment

Reviewer 2 Report

Comments and Suggestions for Authors

Very well presented manuscript, centered on an entity named VVF, which is rare and difficult to manage. The overall presentation, the introduction section and the illustrative cases are presented with details. The discussion section is adequately supported and the presented results are well documented and in accordance with the database of the current literature. Moreover, the number of presented cases is considered as appropriate, given the relative rarity of the underlying disease process.

Comments on the Quality of English Language

Although the quality of English language is good, moderate editing of English language is required.

Author Response

(The authors gave the same response as above.)
